# A Latency-Optimized Network-on-Chip with Rapid Bypass Channels

**DOI:** 10.3390/mi12060621

**Published:** 2021-05-27

**Authors:** Wenheng Ma, Xiyao Gao, Yudi Gao, Ningmei Yu

**Affiliations:** Faculty of Automation and Information Engineering, Xi’an University of Technology, Xi’an 710048, China; wenhma@outlook.com (W.M.); 15829628902@163.com (X.G.); gyd18392005058@163.com (Y.G.)

**Keywords:** network-on-chip, latency optimization, bypass channel, single-cycle multi-hop

## Abstract

Network-on-Chips with simple topologies are widely used due to their scalability and high bandwidth. The transmission latency increases greatly with the number of on-chip nodes. A NoC, called single-cycle multi-hop asynchronous repeated traversal (SMART), is proposed to solve the problem by bypassing intermediate routers. However, the bypass setup request of SMART requires additional pipeline stages and wires. In this paper, we present a NoC with rapid bypass channels that integrates the bypass information into each flit. In the proposed NoC, all the bypass requests are delivered along with flits at the same time reducing the transmission latency. Besides, the bypass request is unicasted in our design instead of broadcasting in SMART leading to a great reduction in wire overhead. We evaluate the NoC in four synthetic traffic patterns. The result shows that the latency of our proposed NoC is 63.54% less than the 1-cycle NoC. Compared to SMART, more than 80% wire overhead and 27% latency are reduced.

## 1. Introduction

Over the last decade, the number of cores and processing elements (PEs) on each chip has greatly increased due to the technology scaling. For complicated digital systems, it was no longer possible to connect hundreds of nodes and transmit data efficiently with conventional networks (buses and crossbars) because of the limitation of their scalability. Instead, Network-on-Chip (NoC), a more flexible connection mechanism, is proposed to meet the performance demand in complex systems. With the help of NoC, networks in digital systems, such as many-core processors, could achieve extremely high bandwidth [1,2,3,4,5]. Inside a NoC, messages traverse from the source node to the destination node through many intermediate routers. As the number of nodes increases, the number of intermediate routers increases greatly, leading to poorer transmission latency. Thus, reducing latency is becoming a key concern for network designers.

For latency reduction, one proposed approach is using very large crossbars [6,7] or high-radix routers with direct connections between remote nodes [8,9,10]. However, these methods require extra ports in routers and additional wire resources between nodes, that are unfriendly to physical design. Owing to the routing complexity of high-radix networks, most system designers have opted for simpler topologies (such as ring or mesh) instead [11,12,13]. For these low-radix networks, reducing the latency of each router is an alternative approach. With aggressive lookahead routing, speculative stages and router bypass mechanisms, transmission delay in such a system has been reduced by optimizing router micro-architecture [14,15,16,17,18,19,20,21,22,23]. Among these approaches, single-cycle multi-hop asynchronous repeated traversal (SMART) network that skips several intermediate routers within a single cycle, achieves a better performance with acceptable area cost [19,20,21,22,23]. Moreover, many researches are proposed to improve SMART, including task mapping, pipeline optimization, etc. [24,25,26]. However, all SMART-based NoCs have to send a bypass request message, called SMART-hop Setup Request (SSR), before transmitting the packet. Additional pipeline stages and wire resources are necessary for SSR delivery, leading to a negative impact on transmission efficiency in many cases.

This paper presents a latency-optimized Network-on-Chip with rapid bypass channels. In the proposed NoC, bypass signals are transmitted with each flit at the same time, rather than delivering them independently in an additional cycle before transmitting the packet. Taking the wire resources into consideration, the bypass request link is shared among all bypass requests by handling bypass conflicts node by node rather than broadcasting the requests and dealing with them together. A fixed priority used in the arbiter and the virtual channel of each packet is allocated statically to reduce the bypass delay. Compared to the SMART NoC [19], more than 80% wires, used for SSR broadcasting, are omitted in our proposed NoC; its latency is 30% less than SMART.

The main contributions of this paper are as follows:We propose to combine the bypass request with flits instead of delivering the request before a packet.A rapid bypass circuit is present, which shows the possibility to transmit the bypass request and the flit simultaneously in an acceptable timing constraint.We evaluate the impact of the startup delay. The result shows that for a single-cycle multi-hop network, lower startup latency could reduce the transmission latency even bypass fewer nodes.

The rest of this paper is organized as follows. Section 2 presents the necessary background and the motivation of the paper. Section 3 provides the architecture and implementation details of the proposed NoC. Wire overhead is evaluated in Section 4. Section 5 shows the performance of the proposed NoC. Finally, Section 6 concludes the paper.

## 2. Related Work

For latency consideration, reducing the transmission delay for each node is a viable methodology. Robert Mullin et al. proposed to remove the routing and arbitration logic from the critical path by precomputing the arbitration decision [14]. In this way, the cycle time is reduced significantly without compromising the router efficiency. Given the pipeline delay, reducing the pipeline stage is also a realistic method for router micro-architecture optimization. In low-load scenarios, many resources in a router are idle, such as output ports, leading to a negative impact on latency. Amit Kumar et al. suggested simplifying the virtual channel allocation which shortens the pipeline [16]. Besides, advanced bundles are used to optimize the pipeline architecture by cutting down the number of pipeline stages in each router at low loads. Hiroki Matsutani et al. implemented a low latency Network-on-Chip with prediction routers [15]. In their proposed NoC, the output channel, used by the next packet, is predicted and the switch allocation is also completed speculatively. For each prediction router, all incoming flits are transmitted without waiting for the routing calculation and switch arbitration when the prediction hits. Owing to the lookahead routing and pre-allocation of the switch and the virtual channel, only two clock cycles are necessary for each hop resulting in a great latency reduction.

Skipping intermediate nodes is an alternative method for latency reduction in addition to optimizing the transmission delay in each node. Express virtual channels (VCs) have been proposed for bypassing intermediate routers between frequently used source–destination pairs reducing the communication latency greatly [17]. It improves the efficiency of packet transmission which spans multiple intermediate nodes. However, it has little impact on those communications between neighboring nodes and the bypass is completed only when enough buffer resource is available. To maximize the utilization of the bypass, Non-Empty Buffer Bypass (NEBB) is proposed [18]. As its name implies, the NEBB NoC allows bypassing a non-empty buffer for single-flit packets or multi-flit packets that there are enough buffers in its destination node. To improve the link utilization, a Single-cycle Multi-hop Asynchronous Repeated Traversal (SMART) Network-on-Chip is presented to build a single-cycle data path all the way from the source to the destination [19,20,21,22,23]. In the SMART NoC, crossbars and links are shared to incoming packets across multi-hops within a single cycle instead of offering any additional fast physical express links in the data path. In this way, flits are allowed to cross multiple nodes in a single cycle reducing the number of average hops.

## 3. Background and Motivation

The architecture of SMART NoC is shown in Figure 1. In each router, there is a bypass channel and an asynchronous repeater for non-stop flit transmitting. Besides, every router has an SSR broadcasting bus connecting to all the other nodes. For each message transmission, a global switching path is established by sending SSR before the packet. Thus, all flits will be transmitted as far as possible until a routing conflict appears. In this way, messages can traverse multiple nodes within a single clock cycle.

However, SMART NoC has three shortcomings: (a) As there are a lot of signals in SSR including the number of hops, virtual net ID, etc., a huge number of additional wires are needed for SSR broadcasting. (b) In order to deal with multiple SSRs at the same time, there must be a big allocator for global switching arbitration. (c) A specific pipeline stage is inserted for SSR delivery. To solve the first two problems, SMART NoC with SSR Net (SMART-SN) is proposed [27]. As demonstrated in Figure 2, a new network is employed for SSR delivery instead of multiple buses. Therefore, an SSR link between adjacent nodes can be shared by all routers. Before the SSR is delivered, an SSR setting signal, called Pre-SSR, is broadcasted to all the other SSR routers setting the global SSR channel. On account of much fewer wires in pre-SSR, the wire overhead of SMART-SN NoC is reduced greatly.

Compared to normal NoCs and the original SMART NoC, the SMART-SN needs more cycles to initiate a packet transmission, due to the additional clock cycle spent on setting SSR network. Figure 3 shows the pipeline of different NoCs. For an optimized conventional NoC (1-cycle NoC), two pipeline stages are employed [28,29]. In SMART NoC, there are three pipeline stages. An individual clock cycle is needed for SSR delivery. As for SMART-SN, another pipeline stage for setting SSR network is necessary, besides the additional cycle for SSR. Owing to the asynchronous repeated traversal, intermediate routers are bypassed directly in both SMART and SMART-SN. However, all the SMART-based NoCs have to spend more cycles on launching a new transmission due to the more complicated pipeline architecture. In other words, a packet may spend more cycles on getting launched in the source node than transmitting it to the destination across all intermediate nodes if there is no conflict. Consequently, speeding up the startup of each transmission in SMART could improve network efficiency. In this paper, we focused on reducing the wire overhead and the launching latency of SMART-based NoC.

## 4. Network-on-Chip with Rapid Bypass Channels

We demonstrate the architecture and the implementation of the proposed NoC in this section.

### 4.1. Architecture

According to the discussion in the last section, SMART-based NoC could benefit from the optimization of the SSR link. To reduce the wire overhead and eliminate the additional pipeline stage, we propose to establish a rapid bypass channel instead of a specific SSR bus or network. Figure 4 shows the architecture of the proposed NoC and the data path inside a router. In our proposed NoC, the bypass request is transmitted along with each flit. Thus, links between routers are the same as normal NoCs. There is no need to employ an additional network or any broadcasting links for bypass requests, which is different from SMART, SMART-SN, and other SMART-based NoCs. For timing consideration, only those intermediate routers along the dimension (X or Y) could be bypassed. All flits, which turn to another direction, must stop at the turn router.

When a flit comes into an intermediate router, the corresponding bypass request arrives at the same time. If a flit arrives at the turn router or there are any flits in the input buffer waiting for transmitting, the flit stops at the router. Moreover, it is stored into the input buffer with its bypass request. If no conflict appears in the current router, the flit will skip to the output port along the dimension through the bypass channel in the same cycle. However, the arbitration logic is too complicated for the bypass channel. To optimize the critical path, we take advantage of two insights.
Packet based flow control. The wormhole mechanism is widely used in different NoCs due to its flexibility [30,31,32], and it is also supported by previous SMART-based NoCs. However, allowing flits of a packet to be blocked and stored in different routers makes it difficult to establish a bypass channel when the bypass request arrives with the flit simultaneously. Thus, we use a packet-based flow control instead. In our proposed NoC, all flits must be ready before the packet leaves the source node. Therefore, transmitting all flits of a packet will not be interrupted. The virtual cut-through flow control is also used for latency reduction [33]. A packet transmission between routers can be done for several consecutive cycles. In this way, a bypass channel keeps working without breaking until the entire transmission of the packet is completed.Static virtual channel. For VC-based NoC, the virtual channel must be decided before delivering the packet to the next router. Allocating VC will cause a huge delay on the critical path. It is unacceptable for flit bypass when the bypass request arrives with flit at the same time. Thus, we use a static VC instead of allocating it dynamically. The VC of a packet is allocated at the source node, and it will be constant during the transmission. In this way, VC allocation in each router is not needed reducing the transmission delay greatly.

Figure 5 shows an example for transmitting three flits in a single cycle. At the beginning of the clock cycle, the status of all routers are shown in Table 1. In N0 and N3, part of the input buffer are occupied by Packet 0 (P0) and Packet 1 (P1), respectively. Meanwhile, Packet 2 (P2) occupies the Local/Turn input buffer of Node 4 (N4) and the Output buffers of N4 and N5. All the other buffers in the network are idle. There are only three packets with different VCs in the network. The details of each packet are as follows.
Packet 0 (P0) is a new packet that will be transmitted to N4 in the following cycles. N0 is the source node of P0. The whole packet is now stored in the local buffer of N0.Packet 1 (P1) is an existing packet that comes from N2. Before getting to the destination (N5), P1 stopped at N3 due to the resource contention in the last cycle. All flits of P1 are stored in the input buffer of N3.Packet 2 (P2) is also an existing packet in the network. Its source node and destination node are N4 and N6 respectively. Part of the flits have already arrived at N6 that is different from P0 and P1. Thus, the output ports of N4 and N5 are occupied by P2.

During the clock cycle, three flits are transmitted forward in the network. Their data paths are shown in Figure 5.
Data path for Packet 0. N0 launches a flit of P0 to N4. The corresponding bypass request is sent to N1, N2 and N3 through the bypass request link in sequence. N1 and N2 are bypassed since all buffers are idle in both of the two nodes. P0 will stop at N3 because P1 is already waiting for transmitting at the input buffer.Data path for Packet 1. N3 launches a flit of P1 to N5. But the flit and the bypass request stop at N4 due to the output port contention caused by P2. In this example, N4 will start transmitting P1 as soon as all flits of P2 leave the node.Data path for Packet 2. At N4, a flit of P2 is launched to N6. It could arrive at the destination node in the same clock without any conflicts.

In this example, three nodes (N1, N2, and N5) are bypassed. As opposed to SMART NoCs, P0 and P1 are launched immediately instead of sending a bypass request before transmitting. P2 could arrive at the destination without breaking owing to the single-cycle multi-hop bypass. In our proposed NoC, the transmission latency is reduced greatly especially for those packets with fewer flits.

### 4.2. Implementation

The micro-architecture and the bypass channel from the west input port to the east output port are demonstrated in Figure 6. There is no VC allocator in the router due to the static VC allocation. A router receives the VC empty signal from the next router, and sends its own VC empty flag to neighbor routers. Flits and their bypass requests get into the router through data link and bypass link at the same time. The bypass request has only two fields: (1) *Hop Left*. It indicates how many hops are left in the current transmission. It is provided by the start node, where the flit is launched, and calculated based on the distance between the start node and the destination node. Besides, it must be smaller than the *Maximum Hops per Cycle* (HPC${}_{m}$), which is a parameter of the network indicating the maximum number of hops for each bypass. If the *Hop Left* is greater than zero, the flit could be transmitted forward skipping the current router. Once a node is bypassed, the *Bypass Request Generator* will decrement the *Hop Left*. (2) *Virtual Channel*. This field records the VC of the flit, which is allocated by the source node.

There are three typical data paths in the router. (1) Store the input packet into the input buffer. If there is any packet waiting for transmitting or the output port is occupied by any other input unit, the arrived flit is stored into the input buffer with its bypass request. Otherwise, the router will start bypass processing immediately. In the bypass channel, the *Bypass Arbiter* will stop the packet if *Hop Left* equals to zero or the requested VC is not empty in the next router. (2) Bypass the input packet. When a bypass channel is established, the flit and the bypass request skip to the output port directly. In the meantime, *Bypass Request Generator* decrements the *Hop Left*. In this way, a single-cycle bypass transmission is finished. Then, the next router will repeat the process to achieve multi-hop bypass. (3) Launch a packet from buffers inside the router. If the packet comes from any buffers of the router, *Route Arbiter* will select a packet and send it to the output port. The new bypass request is also generated according to the header flit of the packet. When the flit and the bypass request arrive at the output port, a new packet transmission begins.

Figure 7 demonstrates the critical path of the bypass channel. When a bypass channel is established, the flit skips to the output port without any processing. It has little impact on timing. As shown in Figure 7, the path for processing bypass requests determines the timing constraint of the entire bypass channel over multiple hops. The widths of *Hop Left* and *Virtual Channel* are log${}_{2}$(HPC${}_{m}$) and N${}_{vc}$, respectively. In most cases, bypassing four or eight routers is enough that we will discuss in the following section. Thus, the *Hop Left* only has two or three bits. Three actions are taken on the bypass request: (1) *Hop Left* decrement. Before sending the new request out, the *Bypass Request Generator* must decrement the *Hop Left*. The subtractor can be optimized as the *Hop Left* has only a few bits. (2) *Hop Left* compare. If the *Hop Left* equals to zero, the *Hop Compare* unit blocks the bypass channel. Considering the width of *Hop Left*, this unit is not on the critical path too. (3) VC check. There are always a lot of VCs in a NoC. These VCs are assigned to different virtual nets. The flit could be bypassed only when the VCs in the current router and the requested VC in the next router are all empty. To finish the VC check as soon as possible, we use a VC vector instead of the VC ID avoiding the unnecessary decoder. Besides, the VC empty of the next router may be blocked according to the status of the input buffer. In this way, the VC empty flags of the current router and the next router are merged together. With the optimizations on the critical path, each bypass processing in a single node could be done in less than 8 levels of logic gates. It is acceptable even for skipping eight nodes in a single clock cycle.

## 5. Wire Overhead

All additional signals for bypass request are shown in Table 2. SSR is necessary in both [19,27]. The wire overhead of each SSR is represented by *N${}_{SSR}$* in (Equation 1). *N${}_{vnet}$* and *N${}_{port}$* represent the number of virtual nets and the number of output ports.
(1)$$\begin{array}{cc} N_{SSR}= & \left\lceil log_{2}\left(1+HPC_{m}\right)\right\rceil +\left\lceil log_{2}\left(N_{vnet}\right)\right\rceil +\\ \; & \left\lceil log_{2}\left(HPC_{m}\right)\right\rceil +\left\lceil log_{2}\left(N_{port}\right)\right\rceil +2 \end{array}$$

SSRs in the SMART NoC are broadcasted to all the other routers including the furthest node. As a result, the number of wires required by a router is the number of bits times *HPC${}_{m}$*. The wire overhead of SMART NoC is
(2)$$W_{SMART}=N_{SSR}\cdot HPC_{m}$$

As opposed to SSR broadcasting in the original SMART NoC, the SMART-SN NoC shares the SSR link between two routers with all SSRs, and uses a Pre-SSR for the SSR network setting. Thus, the wire overhead is the sum of *N${}_{SSR}$* and the source router ID given as
(3)$$W_{SMART\_SN}=N_{SSR}+\left\lceil log_{2}\left(HPC_{m}\right)\right\rceil$$

In our proposed NoC, most of the signals are avoided due to the fixed priority and the static VC. Similar to the SMART-SN NoC, all bypass requests are unicasted. Therefore, the number of bits for bypass request, shown in (Equation 4), is reduced greatly.
(4)$$W_{ThisWork}=\left\lceil log_{2}\left(HPC_{m}\right)\right\rceil +N_{vc}$$

For a mesh network with two virtual nets and two virtual channels per virtual net, the result is shown in Figure 8. The overhead is calculated based on the flit width, which is 128-bit. With the growth of *Maximum Hops per Cycle* (*HPC*${}_{m}$), the wire overhead in SMART increases exponentially. By contrast, there is only a linear increase in SMART-SN and our proposed NoC. The proposed NoC also needs fewer wires than SMART-SN due to the simplified arbitration. For the demo network, the proposed NoC needs 2.34% (*HPC${}_{m}$* = 2) and 4.69% (*HPC${}_{m}$* = 16) more wires than the conventional NoC without additional bypass links. When *HPC${}_{m}$* is 2, the proposed NoC could reduce 83.3% (SMART) and 70% (SMART-SN) wire overhead. When the *HPC${}_{m}$* increases to 16, the wire reduction is 97.5% (SMART) and 68.42% (SMART-SN), respectively.

## 6. Results and Discussion

We use Garnet [34] for our evaluations, which is part of the Gem5 [35,36,37] simulator and provides a cycle-accurate timing model for an on-chip network. The topology of the target network is a 2D-mesh with 8 × 8 nodes. The fixed X-Y routing algorithm is used in the simulation. Each router has 12 virtual channels and a 128-bit flit link. An 1-cycle router is evaluated as a baseline NoC. Moreover, we also simulate the SMART NoC using their open-source code. With the same flit width, the performance of SMART-SN is worse than SMART due to the additional pipeline stage. Thus, we do not evaluate it in our simulation. Four synthetic traffic patterns are used, including Uniform random (send flits to a random destination), Bit complement (send flits to a destination that is the bit complement of the source location), Tornado (send flits to a destination half-way across the X dimension), and Transpose (sends flits to a destination that switches the values of X dimension and Y dimension).

Figure 9 plots the average latency of different NoCs for all synthetic traffic patterns. Compared with the baseline NoC, SMART, and our proposed NoC could reduce the latency greatly due to the multi-hop bypass. Besides, NoCs with specific bypass channels may reach the saturation point at a higher injection rate in both Uniform random traffic and Bit complement traffic. The proposed NoC also consumes fewer cycles than SMART at all injection rates in all traffic patterns. Note that the average latency of the baseline NoC is a function of average hops at a low injection rate. However, our proposed NoC pushes low-load latency to about two to five cycles. For example, the latencies of the baseline NoC are 12.5 (Uniform random), 18.0 (Bit complement), 10.0 (Tornado), and 12.6 (Transpose) when the injection rate is 0.02 flit/node/cycle. While, they are only 3.9 (Uniform random), 4.6 (Bit complement), 2.1 (Tornado), and 4.0 (Transpose) in the proposed NoC.

Latency reduction curves are shown in Figure 10. Compared with the baseline NoC, about 8 to 14 cycles are reduced in low-load scenarios. As the injection rate increases, the number of reduced cycles continues to fall before saturation occurs, as there are more conflicts on the network affecting the bypass efficiency. This decline appears in both SMART and our proposed NoC. In the worst case (in the traffic pattern of Tornado when injection rate is 0.24 flit/node/cycle), the latency reduction has dropped from eight cycles to six cycles. Therefore, it has little impact on the performance of NoCs with bypass channels. Compared with the SMART NoC, the new NoC reduces the transmission latency for each flit. Moreover, the latency reduction increases slightly when injecting more packets into the network. This is caused by the better startup latency which reduces the number of flits on the network leading to less resource contention.

The average latency is shown in Table 3. We only evaluate those results before the network becoming saturation. In all the traffic patterns, our proposed NoC could save 8.52 cycles (Uniform random), 12.59 cycles (Bit complement), 7.38 cycles (Tornado) and 8.30 cycles (Transpose) for each flit. On average, the transmission latency is 5.24 cycles which is 9.2 less than the baseline NoC leading to a 63.54% latency reduction. Even the performance of SMART is much better than the baseline NoC, our proposed NoC still saves 2.4 cycles for each transmission. As the result shows, a lower latency, which is 30% less than that of SMART, is achieved across all synthetic traffic patterns even the latency of SMART is already extremely low.

Furthermore, we study the impact of Max Hops per Cycle (HPC${}_{m}$) on performance. The evaluation results are shown in Figure 11. The suffix after NoC name is used to indicate the number of HPC${}_{m}$. Even only two nodes are allowed to be skipped in a single cycle, the transmission latency of the new NoC, demonstrated by the curve of This Work-2, is much less than the baseline NoC. The latency of the proposed NoC is always lower than SMART when they have the same HPC${}_{m}$. As the injection rate increases, the latency increasing in our proposed NoC is slower than SMART. Besides, a smaller HPC${}_{m}$ also causes less impact due to the shorter pipeline structure. Consequently, the latency of the new NoC (This Work-4), whose HPC${}_{m}$ is only 4, is less than SMART NoC with 8 hops per cycle (SMART-8) in all cases. Even when the HPC${}_{m}$ is two, the latency of the proposed NoC (This Work-2) is similar to SMART-8 in low-load scenarios (injection rate from 0.02 to 0.16) and lower than SMART-8 in other injection rates.

The main shortcomings of the proposed NoC are the utilization of VC buffers and the worse critical path: (1) Limitation of the buffer utilization. As the VC is specified when a packet is launched from the source node, part of the buffer resources in intermediate routers may be wasted in some cases. Fortunately, it has little impact on transmission efficiency due to the extremely low latency. However, more efforts should be made to solve the problem in future work. (2) Longer critical path. Bypass arbitration and flit transmission are completed at the same time in our NoC. Thus, more cells are needed in the bypass path. Although the arbiter is simplified, the critical path is still longer than the original SMART NoC. However, fewer pipeline stages make it possible to achieve lower latency with fewer maximum hops per cycle. Therefore, bypass timing could be optimized by decreasing the maximum bypass distance for each transmission without performance degradation.

## 7. Conclusions

Adding explicit physical channels inside a router to bypass intermediate nodes could achieve lower latency; however, the bypass setup request in previous designs requires additional pipeline stages and a lot of wire resources. In this paper, we present a latency-optimized NoC with rapid bypass channels to reduce the overhead of independent bypass requests. In the new network, all bypass requests are combined with corresponding flits. Moreover, the unicasted mechanism is used instead of broadcasting. The transmission latency of the proposed NoC is 5.24 cycles that is 63.54% less than the 1-cycle NoC. Compared to the SMART NoC, more than 80% of the additional wires are avoided, and a 30% latency reduction is achieved.

This work shows the possibility to transmit the bypass request with flits at the same time. The evaluation result demonstrates that wire overhead is reduced greatly by sharing bypass request links. Besides, the transmission latency of the proposed NoC is less than SMART NoC due to less startup delay. Even if fewer nodes are allowed to be bypassed, our NoC may achieve better performance due to its extremely low latency. It will be helpful to optimize the bypass timing.

## Figures and Tables

**Figure 1 micromachines-12-00621-f001:**
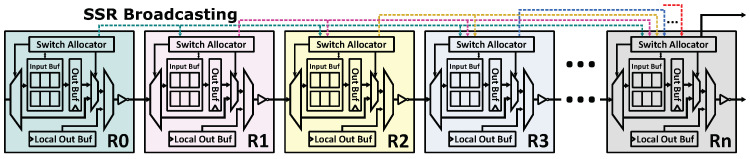
Architecture of SMART NoC.

**Figure 2 micromachines-12-00621-f002:**
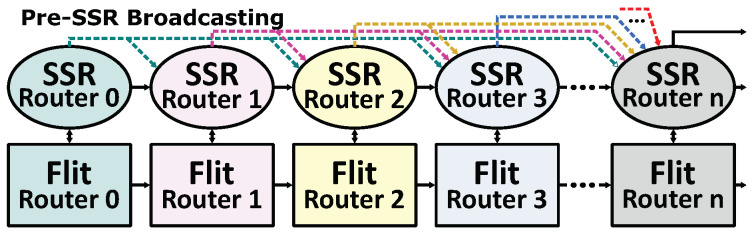
Architecture of SMART NoC with SSR Net.

**Figure 3 micromachines-12-00621-f003:**
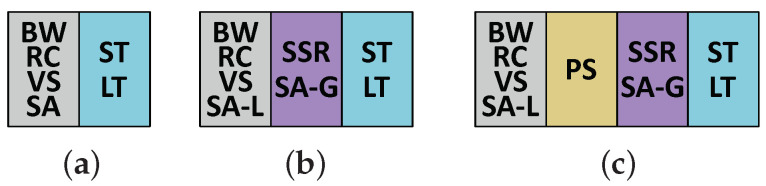
Pipeline of Different NoCs. BW: Buffer Write. RC: Route Compute. VS: Virtual Channel Selection. SA: Switch Allocation. ST: Switch Traversal. LT: Link Traversal. SA-L: Local Switch Allocation. SA-G: Global Switch Allocation. (**a**) 1-cycle NoC, (**b**) SMART NoC, (**c**) SMART NoC with SSR Net.

**Figure 4 micromachines-12-00621-f004:**
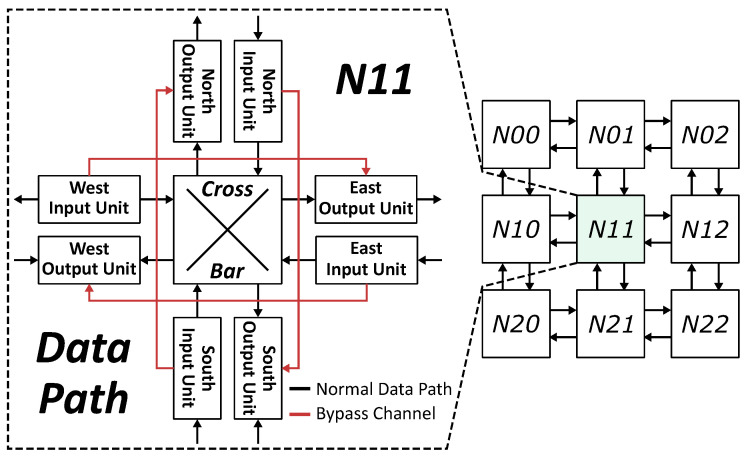
Architecture of the proposed NoC.

**Figure 5 micromachines-12-00621-f005:**
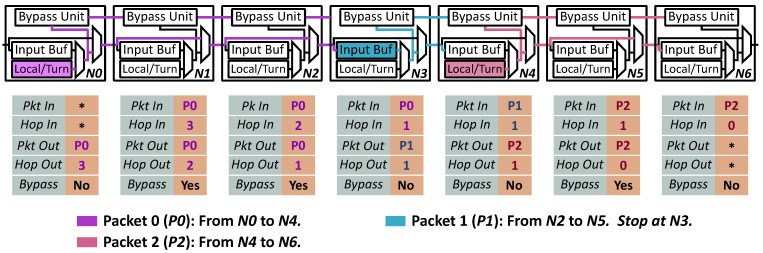
Transmission example.

**Figure 6 micromachines-12-00621-f006:**
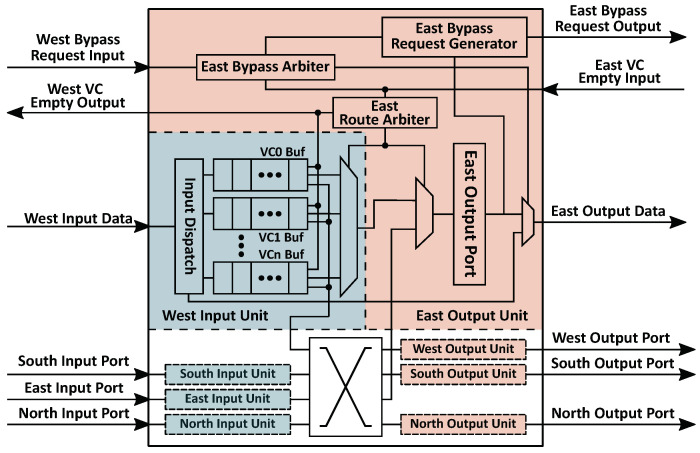
Micro-architecture of the proposed NoC.

**Figure 7 micromachines-12-00621-f007:**
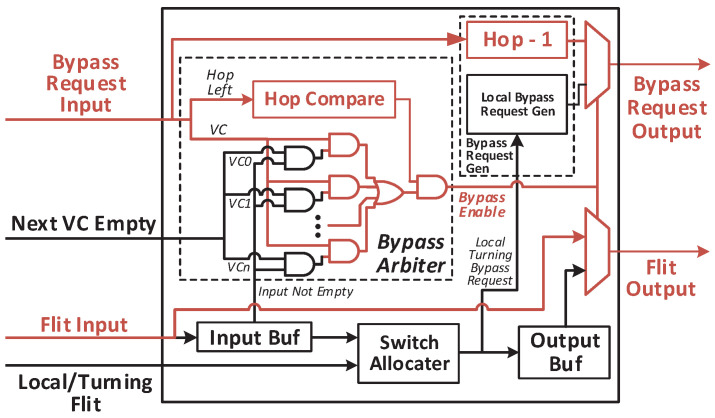
Critical path of the rapid bypass channel.

**Figure 8 micromachines-12-00621-f008:**
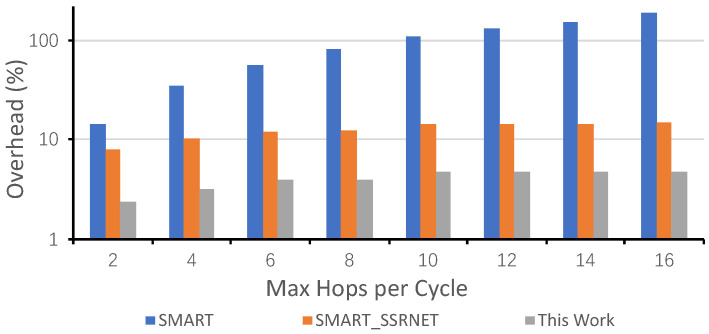
Wire Overhead. The wire overhead is calculated the flit width which is 128-bit in our analysis.

**Figure 9 micromachines-12-00621-f009:**
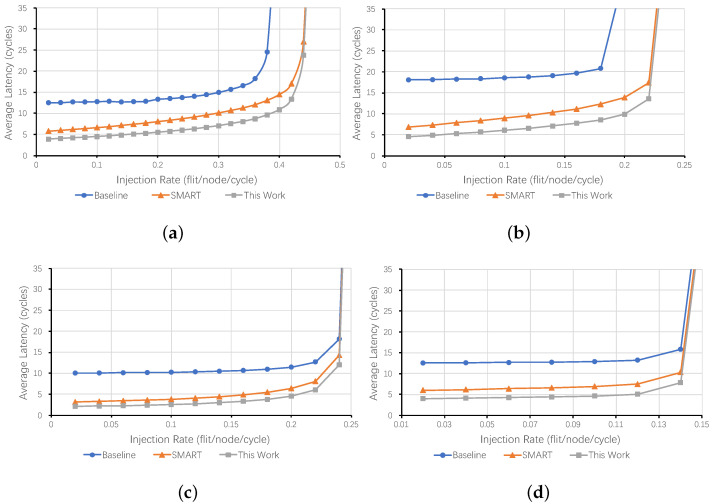
Average Latency in 8 × 8 Mesh Network. (**a**) Uniform Random, (**b**) Bit Complement, (**c**) Tornado, and (**d**) Transpose.

**Figure 10 micromachines-12-00621-f010:**
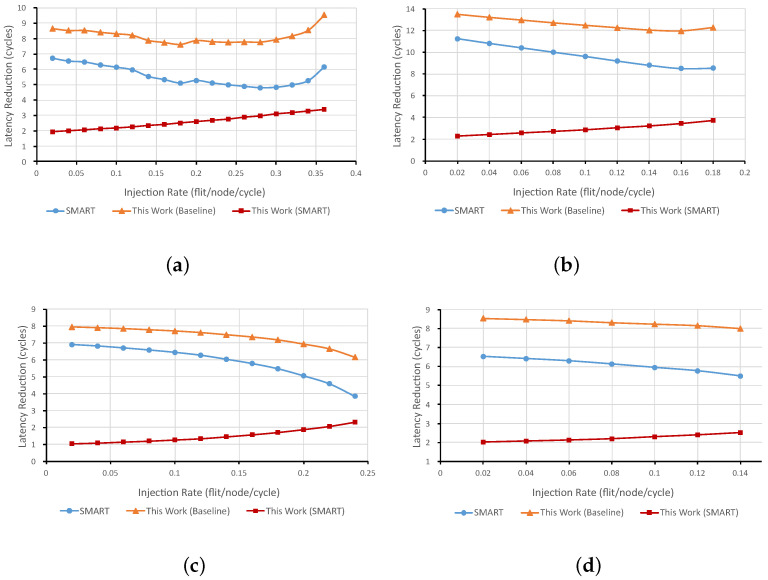
Latency Reduction. (**a**) Uniform Random, (**b**) Bit Complement, (**c**) Tornado, and (**d**) Transpose.

**Figure 11 micromachines-12-00621-f011:**
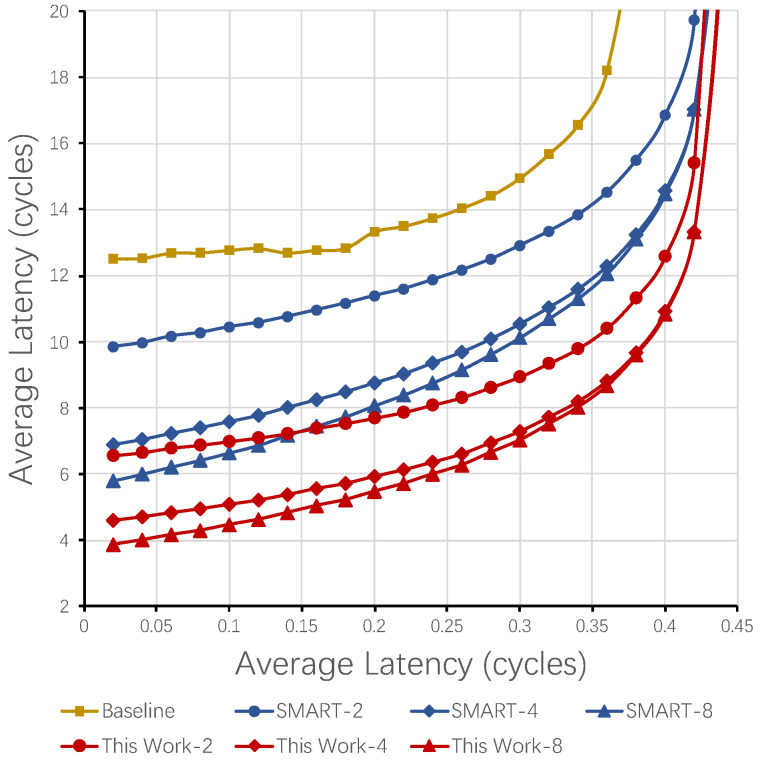
Impact of HPC${}_{m}$.

**Table 1 micromachines-12-00621-t001:** The status of all routers.

Router	Input Buffer along Dimension	Local/Turn Input Buffer	Output Buffer
N0	Packet 0	/	/
N1	/	/	/
N2	/	/	/
N3	Packet 1	/	/
N4	/	Packet 2	Packet 2
N5	/	/	Packet 2
N6	/	/	/

**Table 2 micromachines-12-00621-t002:** SSR signal list.

	Signal	Width	Description
SSR in [19,27]	hop_num	log${}_{2}$ (1 + HPC${}_{m}$)	Number of hops.
	vnet_id	log${}_{2}$ (N${}_{vnet}$)	Virtual network ID.
	inject_router_id	log${}_{2}$ (HPC${}_{m}$)	Source router ID.
	head_flit_flag	1	Header flit flag.
	eject_flag	1	Arrive at the destination.
	eject_port_id	log${}_{2}$ (N${}_{port}$)	Ejection port at destination.
Pre-SSR in [27]	source_id	log${}_{2}$ (HPC${}_{m}$)	Source router ID
Rapid bypass in this work	hop_left	log${}_{2}$ (HPC${}_{m}$)	Hops left over.
	vc	N${}_{vc}$	Virtual channel ID.

**Table 3 micromachines-12-00621-t003:** Average latency of Proposed NoC, Baseline 1-cycle NoC and SMART.

		Uniform Random	Bit Complement	Tornado	Transpose	Average
This Work vs. Baseline	Injection Rate	0.02∼0.38	0.02∼0.18	0.02∼0.24	0.02∼0.14	/
	Baseline (cycles)	14.38	18.84	11.29	13.24	14.44
	This Work (cycles)	5.86	6.25	3.91	4.93	5.24
	Reduction (cycles)	8.52	12.59	7.38	8.30	9.20
	Reduction (%)	59.25%	66.81%	65.38%	62.72%	63.54%
This Work vs. SMART	Injection Rate	0.02∼0.44	0.02∼0.22	0.02∼0.24	0.02∼0.14	/
	SMART (cycles)	10.00	10.34	5.41	7.16	8.23
	This Work (cycles)	7.24	7.25	3.91	4.93	5.83
	Reduction (cycles)	2.76	3.10	1.51	2.23	2.4
	Reduction (%)	27.62%	29.94%	27.81%	31.10%	29.12%

## Data Availability

No new data were created or analyzed in this study. Data sharing is not applicable to this article.

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
