# Peer review of "A Latency-Optimized Network-on-Chip with Rapid Bypass Channels"

_micromachines, 2021, doi:10.3390/mi12060621_

Round 1

Reviewer 1 Report

The paper is well-written, scientifically sound, and proposes a novel solution - as the title truly suggests, the main idea is the latency-optimized bypass channel for NoCs.

The main contributions of the authors are threefold; they describe these well in the paper. First, they propose to combine the bypass request with flits instead of delivering the request before a packet. Second, a rapid bypass is proposed, showing the possibility to transmit the bypass request and the flit simultaneously in an acceptable timing constraint. Third, they evaluate the impact of the startup delay, and the presented results are convincing.

There are two minor issues with the citations. First, there should be a space before the "[" brackets - throughout the paper. Second, aggregated citations such as [14-23] are not that convincing - at least the main achievements, definitions, or novelties of those individual papers should be mentioned. Maybe a separate Related Works section would be beneficial.

Minors:

-- Figure 3: Not all the abbreviations are listed in the caption. It would make the reader's comprehension better if all abbreviations appeared.
-- Figure 8: Wire overhead is represented as %, so the caption should mention what is the base (e.g. 100%) of the calculation. This should be clarified in the text.

The possible drawbacks of the solution should be described in the main text as well as in the Conclusions. Altogether the current conclusion seems like a short summary; the main achievements and their effect should be elaborated better.

Author Response

Dear reviewer,

According to your suggestion, there are 4 modifications.

  1. We write a new section for related works. It is shown in section 2 (line 65).
  2. Figure 3 and Figure 8 are updated.
  3. We discuss the drawbacks in section 6 (line 331).
  4. The conclusion is updated.

If any other modifications are needed, please tell me. It's my pleasure to improve my paper according to your suggestions.

Reviewer 2 Report

In this work, the authors discuss the problem of transmission latency in Network-on-Chip (NoC). The authors propose a mechanism that reduces the latency by bypassing intermediate nodes.

It is a well-written paper, well-structured and organized. The authors support their works with clear pictures and experiments, that were well conducted, and their analysis was well performed.

I think it is a good work.

Author Response

Dear reviewer,

We write a new section for related works. It is shown in section 2 (line 65). Besides, drawbacks are discussed in section 6 (line 331).

If any other modifications are needed, please tell me. It's my pleasure to improve my paper according to your suggestions.
